# Varicella-zoster virus VLT-ORF63 fusion transcript induces broad viral gene expression during reactivation from neuronal latency

Werner J. D. Ouwendijk [1,7], Daniel P. Depledge[2,7], Labchan Rajbhandari [3], Tihana Lenac Rovis[4], Stipan Jonjic [4], Judith Breuer [5], Arun Venkatesan[3], Georges M. G. M. Verjans[1] & Tomohiko Sadaoka [6✉]

Varicella-zoster virus (VZV) establishes lifelong neuronal latency in most humans world-wide, reactivating in one-third to cause herpes zoster and occasionally chronic pain. How VZV establishes, maintains and reactivates from latency is largely unknown. VZV transcription during latency is restricted to the latency-associated transcript (VLT) and RNA 63 (encoding ORF63) in naturally VZV-infected human trigeminal ganglia (TG). While significantly more abundant, VLT levels positively correlated with RNA 63 suggesting co-regulated transcription during latency. Here, we identify VLT-ORF63 fusion transcripts and confirm VLT-ORF63, but not RNA 63, expression in human TG neurons. During in vitro latency, VLT is transcribed, whereas VLT-ORF63 expression is induced by reactivation stimuli. One isoform of VLT-ORF63, encoding a fusion protein combining VLT and ORF63 proteins, induces broad viral gene transcription. Collectively, our findings show that VZV expresses a unique set of VLT-ORF63 transcripts, potentially involved in the transition from latency to lytic VZV infection.

[1] Department of Viroscience, Erasmus Medical Centre, 3015 CN Rotterdam, The Netherlands. [2] Department of Medicine, New York University School of Medicine, New York, NY 10016, USA. [3] Division of Neuroimmunology and Neuroinfectious Diseases, Department of Neurology, Johns Hopkins University School of Medicine, 600 N. Wolfe St., Meyer 6-113, Baltimore, MD 21287, USA. [4] Center for Proteomics, Faculty of Medicine, University of Rijeka, Rijeka 51000, Croatia. [5] Division of Infection and Immunity, University College London, London WC1E 6BT, UK. [6] Division of Clinical Virology, Center for Infectious Diseases, Kobe University Graduate School of Medicine, 7-5-1 Kusunoki-cho, Chuo-ku, Kobe 650-0017, Japan. [7] These authors contributed equally: Werner J. D. Ouwendijk, Daniel P. Depledge. ✉email: tomsada@crystal.kobe-u.ac.jp

The ubiquitous human neurotropic alphaherpesvirus (αHV) varicella-zoster virus (VZV) establishes lifelong latency in sensory neurons of dorsal root and cranial nerve ganglia, as well as autonomic and enteric ganglia[1,2]. VZV reactivates in about one-third of latently infected individuals to cause herpes zoster (HZ), a debilitating disease often complicated by post-herpetic neuralgia (PHN)[3,4]. The incidence and severity of HZ are closely related to declining VZV-specific T-cell immunity by natural senescence[5], and immunosuppressive diseases[6]. However, the mechanisms governing how VZV reestablishes a lytic infection from latency in neurons, especially the identity of the viral gene(s) that initiate reactivation in response to cellular signaling, remain unknown.

During latency in human trigeminal ganglia (TG), VZV gene expression is restricted to transcripts arising from VLT (VZV latency-associated transcript) and ORF63 loci[7]. VLT is a polyadenylated RNA comprising five exons that lie partially antisense to VZV ORF61, the infected cell polypeptide 0 (ICP0) homologue conserved among αHV. Based on genomic location and expression pattern, VLT is considered a homologue of the latency-associated transcripts (LATs) encoded by all other well-studied neurotropic αHVs[8]. While VLT is the most prevalent and abundant VZV transcript expressed in human TGs, lower levels of ORF63 RNA have also been reported in up to 70% of examined latently VZV-infected ganglia[7,9]. This apparent expression of two distinct viral transcripts during latency is unique among well-studied αHV. Expression levels of VLT and ORF63 transcripts correlate significantly[7], suggesting co-regulated expression of both transcripts during in vivo latency, and that these transcripts and/or their encoded proteins may play an important role in the VZV latency and reactivation cycle. VLT is expressed during both latent and lytic VZV infection[7]. Intriguingly, VLT isoforms expressed during lytic VZV infection ($_{lyt}$VLT) differ extensively from the latent VLT isoform in that they contain additional upstream exons and show evidence of alternative splicing by exon skipping or intron retention[7]. Furthermore, a long-read cDNA sequencing approach recently reported the presence of $_{lyt}$VLT splicing into the ORF63 transcripts during lytic VZV infection[10], raising the possibility that individual transcripts may span the VLT and ORF63 loci.

Here, we set out to characterize the expression patterns and functional importance of VLT and ORF63 encoding RNAs during lytic and latent infections. We apply nanopore direct RNA sequencing (dRNA-Seq)[11] to examine lytically VZV-infected epithelial cells and discover a novel set of VLT-ORF63 fusion transcripts, which are also present in latently VZV-infected human TG and our recently improved in vitro VZV human neuronal latency model based on human induced pluripotent stem cell (iPSC)-derived sensory neurons (HSN)[7,12].

## Results

### Identification of multiple VLT-ORF63 fusion transcripts in lytically VZV-infected cells.
Sequencing of full-length RNAs, including direct RNA sequencing (dRNA-Seq), is particularly useful for disentangling complex loci at which multiple transcript isoforms overlap[13,14]. Recently, we used dRNA-Seq to reannotate the lytic VZV transcriptome and redefined the kinetic classes of all canonical and newly identified transcript isoforms[15]. Here, we performed more detailed dRNA-Seq analysis of the VLT and ORF63 loci in lytically VZV-infected human retina epithelial (ARPE-19) cells, identifying multiple lytic VLT ($_{lyt}$VLT) isoforms, all of which contain one or more upstream exons to the core VLT region (exons 1–5) that defines the latent isoform, referred to as VLT[7]. By far the most common $_{lyt}$VLT variants utilized a single upstream exon designated as exon A (Fig. 1a). We additionally

identified several relatively abundant transcripts that spanned the VLT and ORF63 loci, $_{lyt}$VLT-ORF63 fusion transcripts hereafter referred to as $_{lyt}$VLT63. Two major $_{lyt}$VLT63 isoforms (i.e. $_{lyt}$VLT63-1 and $_{lyt}$VLT63-2) represent alternatively spliced variants of a lytic VLT isoform comprising the VLT core region, the additional upstream exon A, and the RNA 63-1 encoding canonical ORF63[15]. $_{lyt}$VLT63-1 and $_{lyt}$VLT63-2 differ from each other by skipping or retaining VLT exon 5, respectively. Both isoforms use a splice acceptor site located 71 nucleotides (nt) upstream of the ORF63 coding sequence (CDS), located within the 5′-untranslated region (UTR) of canonical ORF63[15–17]. A third major isoform, $_{lyt}$VLT63-3 utilizes a unique transcription start site (TSS), not used by any of the other $_{lyt}$VLT and $_{lyt}$VLT63 isoforms, proximal to exon 5 and is predicted to encode pORF63 (Fig. 1a).

To investigate the effect of cell type on lytic transcription across the VLT and ORF63 loci, we assayed our in vitro HSN model that supports both lytic and latent VZV infection[12]. The low yields of viral RNA obtained from infected HSN cultures necessitated the use of nanopore cDNA sequencing (cDNA-Seq) rather than dRNA-Seq. Notably, the same $_{lyt}$VLT and $_{lyt}$VLT63 isoforms were detected in lytically VZV-infected HSN and ARPE-19 cells by both nanopore sequencing (Fig. 1a) and RT-qPCR (Fig. 1b) using primer sets spanning $_{lyt}$VLT63 isoform-defining exons (Supplementary Fig. 1 and Supplementary Table 1). Collectively, these data indicate that identical repertoires of $_{lyt}$VLT and $_{lyt}$VLT63 isoforms are expressed during lytic VZV infection in human sensory neurons and epithelial cells.

### VLT-ORF63 explains co-regulated VLT and RNA 63 transcription in latently VZV-infected human TG.
RT-qPCR was performed on four human TG specimens to determine whether viral RNA detected in latently VZV-infected TG should be ascribed to the pORF63 encoding RNA 63-1 transcript and/or the VLT-ORF63 fusion transcripts VLT63-1 and VLT63-2 (Supplementary Table 2). Consistent with our earlier study[7], VLT but not $_{lyt}$VLT (i.e. no detectable presence of exon A) was detected in all TG analyzed, while the ORF63 CDS region of RNA 63-1 was detected in 3 of 4 specimens (Fig. 2a). Similarly, splice junction usage between VLT exon 4 and RNA 63-1, and exon 5 and RNA 63-1 was detected in all three TGs positive for the ORF63 CDS region (Fig. 2a). No transcripts were detected using a primer set targeting the 5′ UTR region of canonical RNA 63-1 that is absent in all VLT-ORF63 transcripts (Supplementary Fig. 1). These results imply that VLT and VLT63 isoforms, but not $_{lyt}$VLT, $_{lyt}$VLT63 isofoms, nor RNA 63-1 are expressed in latently VZV-infected human TG.

We next performed multiplex fluorescent in situ hybridization (mFISH) using probes directed to the VLT and ORF63 CDS to simultaneously profile VLT, VLT63-1, VLT63-2, and RNA 63-1 expression in the same TG section. During lytic infection in HZ skin biopsies (Fig. 2b, lower panels) and ARPE-19 cells (Supplementary Fig. 2a), VLT and ORF63 CDS ISH signal showed partial co-localization within the nucleus and largely divergent localization within the cytoplasm of VZV-infected cells. Analysis of seven additional human TGs (Supplementary Table 3) confirmed nuclear VLT RNA (Fig. 2b, middle panels) or co-localization of nuclear VLT and ORF63 CDS ISH signals (Fig. 2b, upper panels) in VZV-infected neurons. Notably, the two ISH signals co-localized in the vast majority of neurons (i.e. 4-11 neurons/section, present in all 7 TGs analyzed). Only a small number of neurons, 4 neurons in total and in just 2 of 7 TGs analyzed, yielded staining that showed distinct puncta (Supplementary Fig. 2b). To exclude VZV reactivation in neurons expressing VLT63 we performed mFISH using probes directed to

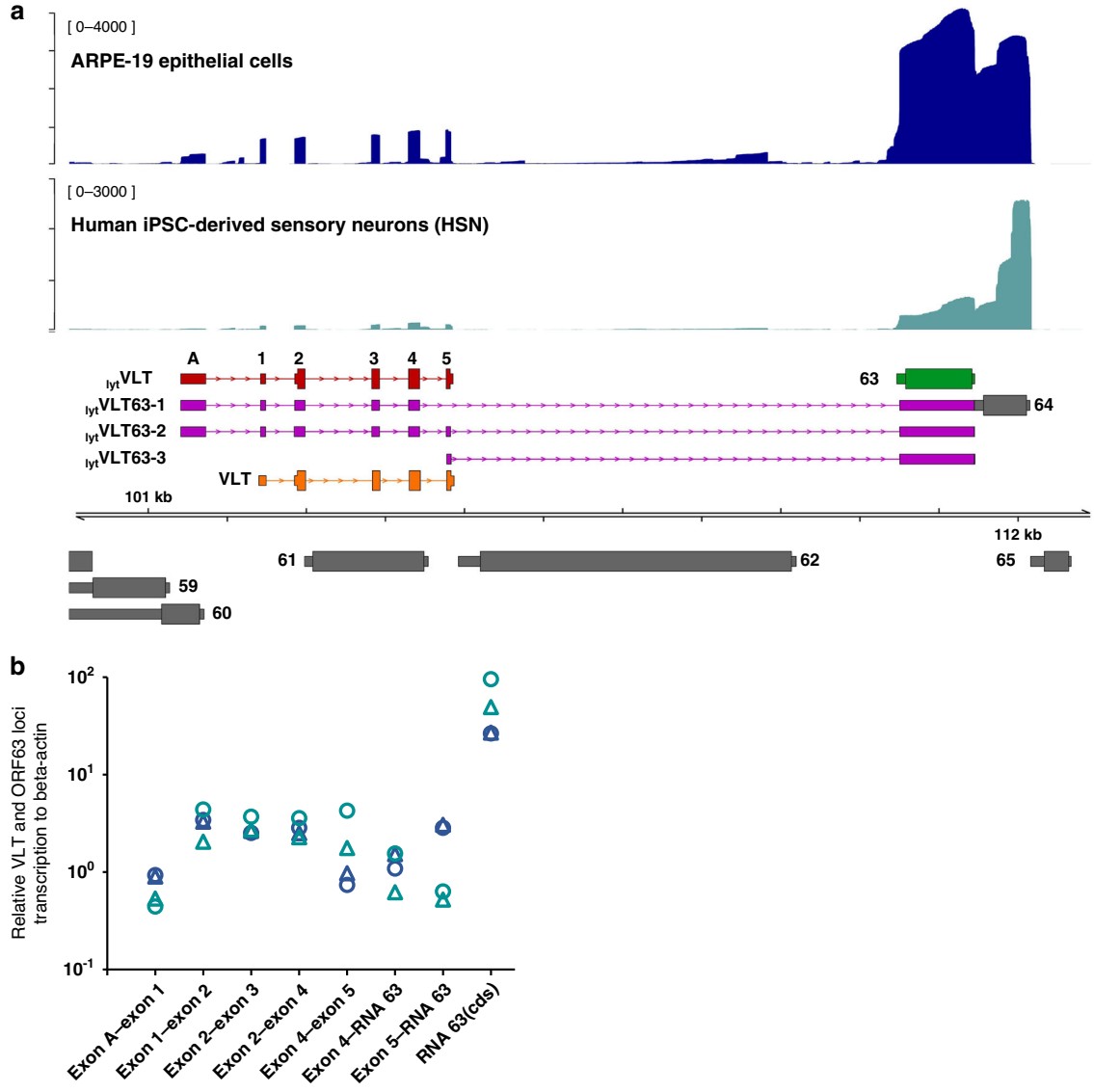

**Fig. 1 Transcription profile across the VLT and ORF63 loci during lytic VZV infection of human epithelial cells and sensory neurons. a** Coverage plots denoting dRNA-Seq (ARPE-19 epithelial cells, dark blue) and cDNA-Seq (human iPSC-derived sensory neurons [HSN], teal) data aligned to the top strand of the VZV genome. dRNA-Seq data is representative of two independently sequenced biological replicates with RNA extracted from lytically VZV-infected ARPE-19 cells at 4 days post infection (4 dpi). cDNA-Seq data was generated from a pool of biological replicates collected from lytically VZV-infected HSN at 14 dpi. Schematics of the major transcripts from VLT and ORF63 loci are shown in following colors: lytic VLT isoform ($_{lyt}$VLT, red), canonical RNA 63-1 (63, green), lytic VLT-ORF63 isoforms ($_{lyt}$VLT63-1/2/3, purple) and latent VLT isoform (VLT, orange). Additional canonical VZV transcription units present are shown in grey, with ORF numbers indicated. Wide boxes indicate canonical CDS domains and while thin boxes indicate UTRs, respectively. Y-axis values indicate the maximum read depth of that track. **b** Analysis of VLT, RNA 63 and VLT-ORF63 isoform expression by RT-qPCR analysis using the same two independent experiments in ARPE-19 cells (dark blue) and HSN (teal) for long-read sequencing. The primer locations used for RT-qPCR analysis detecting transcripts from VLT to ORF63 loci are depicted in Supplementary Fig. 1. Source data are provided as a Source Data file. ORF; open reading frame, CDS; coding sequence, UTR; untranslated region.

VLT, the ORF63 CDS, and a region of the ORF9 loci which encodes multiple transcripts including RNA 9-1, the most abundantly lytic transcript. Here, strong ISH signal for VLT and ORF63 CDS was observed in VZV-infected epithelial cells present in both HZ skin biopsies (Supplementary Fig. 3a) and latently infected neurons, whereas ISH signal for RNA 9 was only observed in the skin biopsies (i.e. no RNA 9 transcripts present in latently infected neurons). Thus, co-localization of VLT and ORF63 CDS ISH signal during latency provides further support for the presence of the VLT63-1 and VLT63-2 fusion transcripts in latently VZV-infected human TG.

To further characterize and differentiate VLT, VLT63-1, VLT63-2, and RNA 63-1 transcripts in human TGs, 5′-RACE analysis was performed on pooled poly(A)-selected RNA collected from 3 TGs with detectable VLT-ORF63 fusion transcripts (i.e. TG 2–4 in Fig. 2a). The VLT-specific reverse primers (VLTexon4R104361 and VLTexon5R104799 in Fig. 2c and Supplementary Table 4) identified two alternative TSS located 4 nt upstream and 21 nt downstream of VLT exon 1 (Fig. 2c, orange arrows in top row), and detected only VLT with no alternatively spliced isoforms (Fig. 2c, orange boxes). The primers specifically binding to the ORF63 CDS (ORF63R622 and

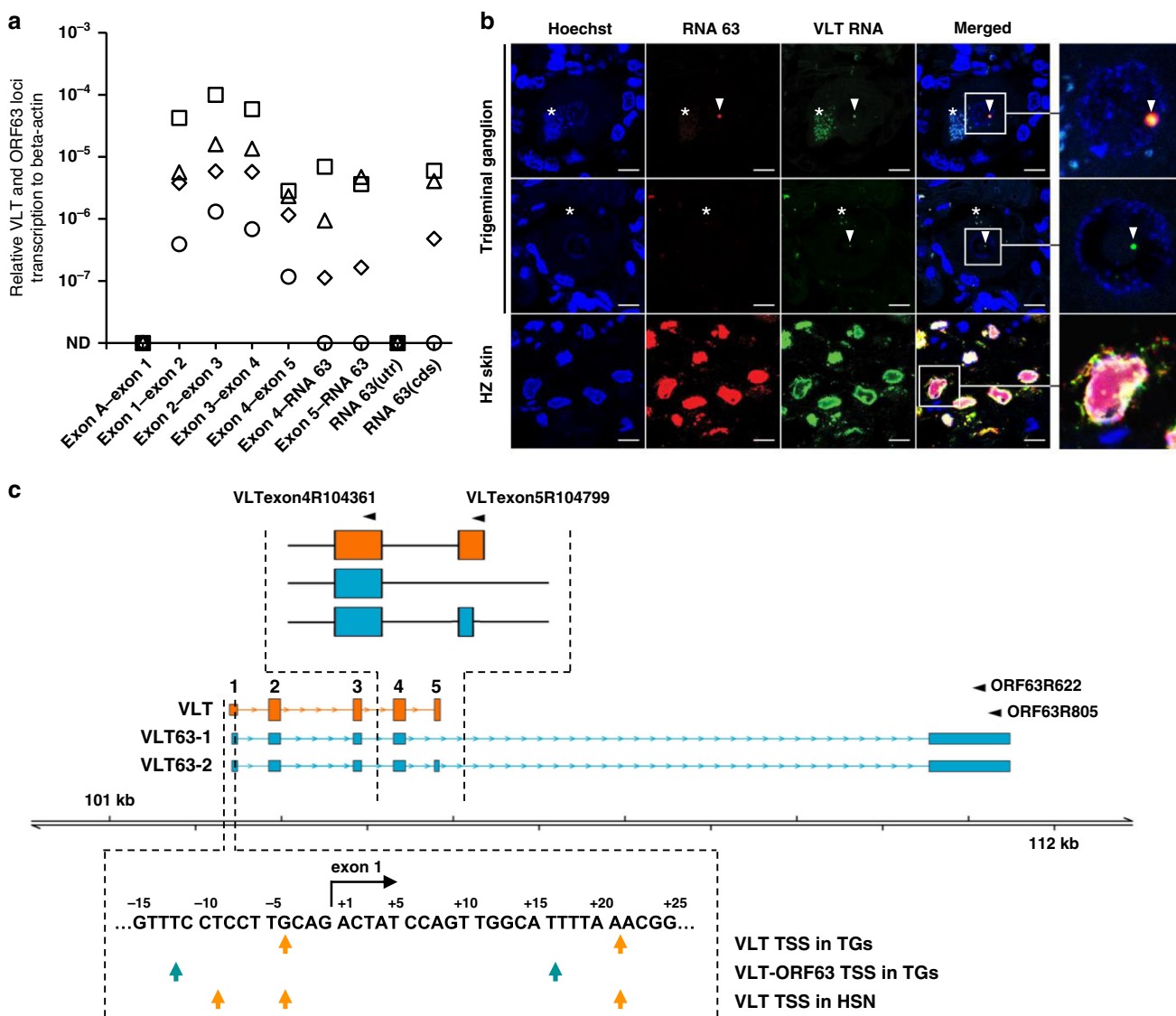

**Fig. 2 Transcription profile across the VLT and ORF63 loci in latently VZV-infected human trigeminal ganglia. a** Detection of VLT, canonical RNA 63-1 and VLT63 isoforms by RT-qPCR analysis in latently VZV-infected human trigeminal ganglia (TG) ($n = 4$; post-mortem interval 4–4.5 h). Data on individual TG samples are shown as unique symbols. Source data are provided as a Source Data file. **b** Detection of VZV RNA 63 and VLT RNA by multiplex fluorescent in situ hybridization (ISH) on human TG (upper two panels) and human herpes zoster skin biopsy (bottom panel). Representative images are shown for $n = 7$ human TG and $n = 2$ zoster skin biopsies. Asterisks indicate autofluorescent lipofuscin granules in neurons. Scale bar: 10 μm. Arrowheads indicate RNA 63 (red) and/or VLT (green) ISH signal. Right panels: enlargements of area indicated by white box. **c** Putative transcription start sites (TSS) of VLT (row 1), VLT63 (row 2) in human TGs and VLT in latently VZV-infected human iPSC-derived sensory neurons (HSN) (row 3), as determined by 5′-RACE analysis. Schematic top shows major latent VLT and VLT-ORF63 transcript isoforms and location of primers used for 5′-RACE analysis. Bottom: VLT sequence of VLT exon 1, as previously determined by RNA-seq on human TGs[7]. Flanking regions are shown with arrows indicating putative TSS by 5′-RACE analysis.

ORF63R805 in Fig. 2c and Supplementary Table 4) identified only two VLT-ORF63 isoforms, VLT63-1 and VLT63-2 with alternative splicing donor sites located in VLT exons 4 and 5, respectively (Fig. 2c, teal boxes). Note that VLT63-1 and VLT63-2 are identical to $_{lyt}$VLT63-1 and $_{lyt}$VLT63-2 respectively expressed during lytic VZV infection, except for the unique 5′ ends that discriminate latent isoform from $_{lyt}$VLT variants (Fig. 1a). Importantly, the TSS for all RNAs containing RNA 63-1 sequence in human TG were located close to VLT exon 1 and were part of the VLT-ORF63 fusion transcripts (Fig. 2c, teal arrows), while no canonical RNA 63-1 transcripts were detected. Collectively, these results implicate that most – if not all – RNA 63-1 previously detected in latently VZV-infected human TG is attributed to VLT63-1 and VLT63-2 transcript expression.

**Protein coding potential differs between VLT-ORF63 transcript isoforms.** In silico translation of the major lytic VLT-ORF63 isoforms, $_{lyt}$VLT63-1 and $_{lyt}$VLT63-2, predicted novel proteins (Fig. 3a and Supplementary Fig. 4), whereas $_{lyt}$VLT63-3 (Fig. 1a) appears to encode solely canonical ORF63 protein (pORF63). Importantly, exon A utilized in lytic isoforms does not affect the protein coding potential, suggesting that the respective latent and lytic isoforms encode the same protein i.e. VLT and $_{lyt}$VLT are both predicted to encode identical version of pVLT (Fig. 3a). VLT63-1 ($_{lyt}$VLT63-1) encodes a putative in-frame fusion protein called pVLT-ORF63. The pVLT-ORF63 is predicted to be 421 amino acids (aa) in length, comprising partial pVLT (aa 1-119) and the canonical pORF63 (278 aa) linked together by 24 aa polypeptide (GFVRFITRQRRVGFKGKGYYGPKD) encoded within the partial

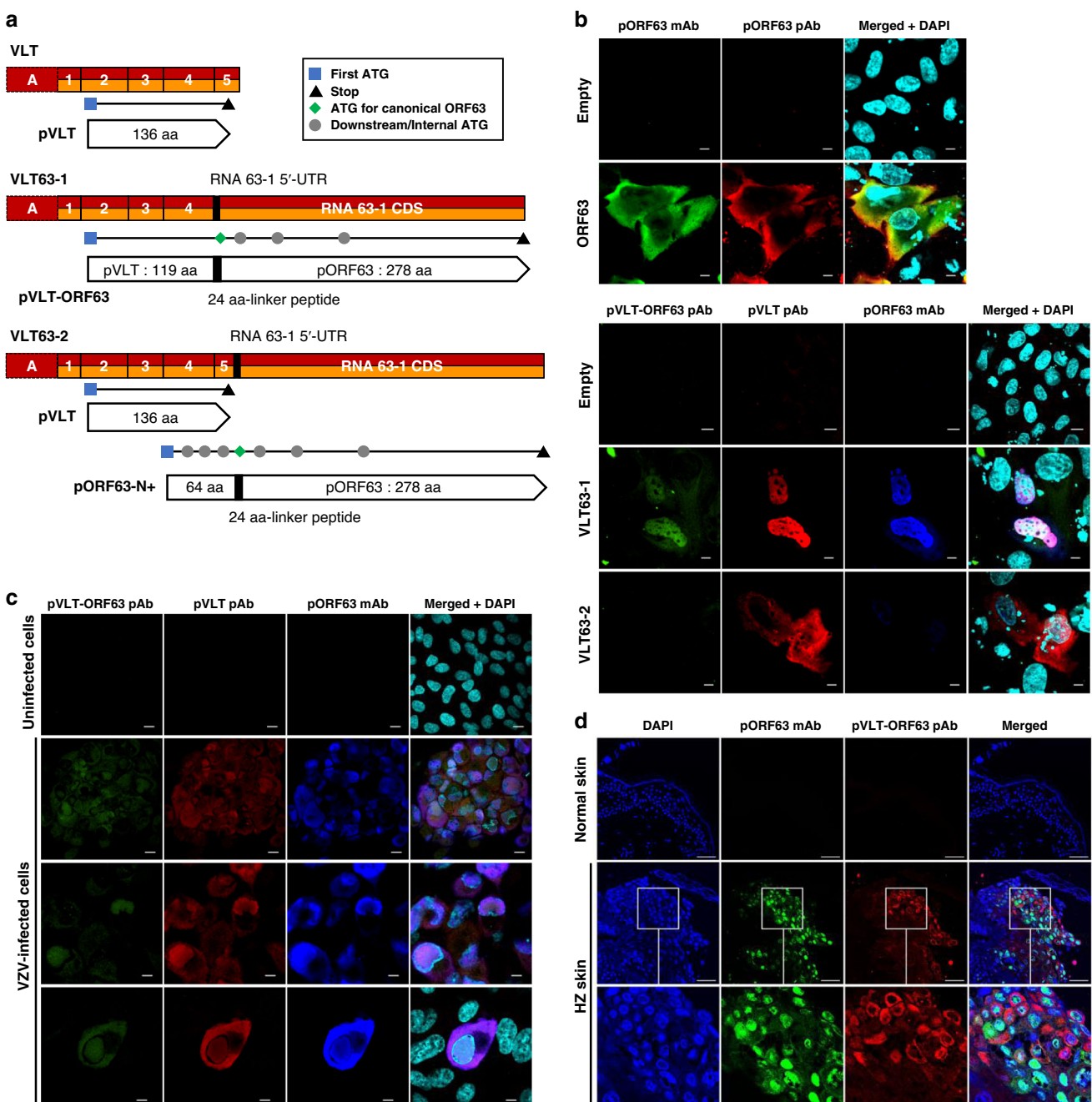

**Fig. 3 Protein coding potential of VLT-ORF63 fusion transcripts. a** Schematic presentation of VLT and VLT-ORF63 isoform transcripts with predicted encoded proteins. Red boxes indicate exons of lytic isoforms of VLT and VLT-ORF63 transcripts, orange boxes indicate exons of latent isoforms of VLT and VLT-ORF63 transcripts, and a black box indicates a part of RNA 63-1 5′-UTR in canonical RNA 63-1 transcript. The end of VLT-ORF63 transcripts indicates stop codon for ORF63 CDS. Black horizontal lines indicate location of encoded open reading frames (ORFs). The blue square indicates the first start codon (ATG), the black triangle indicates first stop codon within the same frame as the start codon, the green diamond indicates start codon of canonical ORF63, and grey circles indicate downstream ATG codons within the same frame as the start codon. Pointed rectangles show translated protein from corresponding ORFs with a black box indicating the 24 amino acid linker peptides translated from a part of RNA 63-1 5′-UTR in the canonical RNA 63-1 transcript. UTR; untranslated region, CDS; coding sequence. Confocal microscopic images of ARPE-19 cells **b** transfected with CS-CA-VZV plasmids (48 h post transfection), **c** lytically infected with VZV (4 dpi), and **d** herpes zoster skin lesions. **b, c** Cells were stained with anti-pORF63 mAb (green) and anti-pORF63 pAb (red) for CS-CA-ORF63, and anti-pVLT-ORF63 pAb (green), anti-pVLT pAb (red) and anti-pORF63 mAb (blue) for CS-CA-VLT-ORF63-1 and CS-CA-VLT-ORF63-2 and lytic VZV infection. Nuclei were stained with DAPI (cyan) and images are representative of results from two independent experiments. Magnification; x600 and x3 digital zoom with 5 μm scale bars (except 3rd row in **b**, and 3rd and 4th rows in **c**) and x600 and x2 digital zoom with 10 μm scale bars (3rd row in **b**, and 1st and 2nd rows in **c**). mAb; monoclonal antibody, pAb; polyclonal antibody. **d** Nuclei were stained with DAPI (blue) and images are representative for two independent stainings performed on one control and two herpes zoster skin biopsies. Magnification: x200 with 50 μm scale bars (1st and 2nd rows) and x200 and x3 digital zoom with 20 μm scale bars (3rd row).

5′-UTR of RNA 63-1 (Fig. 3a). The VLT63-2 ($_{lyt}$VLT63-2) isoform is predicted to encode two separate proteins: complete pVLT (136 aa) and an N-terminally extended 336 aa pORF63 protein variant (pORF63-N+). pORF63-N+ includes an N-terminal 88 aa polypeptide, the first 64 aa of which are translated from VLT exons 4 and 5, but out-of-frame with pVLT, and the same 24 aa linker polypeptide of pVLT-ORF63 encoded from the partial 5′-UTR of RNA 63-1 (Fig. 3a).

The protein coding capacity of the VLT-ORF63 isoforms were examined by immunoblotting of ARPE-19 cells transfected with plasmids expressing VLT63-1 and VLT63-2, and ORF63 for comparison, as well as in lytically VZV-infected ARPE-19 cells using antibodies raised against parts of pVLT and pORF63 (Supplementary Fig. 5). The anti-pVLT antibody (Ab) was raised against the first 19 aa of pVLT[7], being part of both pVLT and pVLT-ORF63. The anti-pORF63 Ab was raised against whole pORF63[7], recognizing pVLT-ORF63, pORF63-N+ and pORF63. Translation of all predicted proteins from each plasmid was confirmed in transfected ARPE-19 cells and both pVLT-ORF63 and pORF63-N+ were readily detected in the context of lytic VZV infection (Supplementary Fig. 5).

Next, we assayed the cellular localization of protein expression by immunofluorescence. In transfected cells, pORF63 was detected in the nucleus and particularly in the cytoplasm, whereas pVLT-ORF63 predominantly localized in the nucleus (Fig. 3b). Cells transfected with the VLT63-2 vector, predicted to encode pVLT and the fusion protein pORF63-N+, resulted in nuclear and cytoplasmic pVLT but undetectable pORF63-N+ expression (Fig. 3b). In lytically VZV-infected ARPE-19 cells, pVLT-ORF63 localized to the nucleus and cytoplasm, as demonstrated by immunofluorescent staining with antibodies against pVLT-ORF63, pVLT and pORF63 (Fig. 3c), possibly indicating that other viral proteins are required for its retention in cytoplasm or shuttling between nuclear and cytoplasm. Finally, we confirmed expression of pVLT-ORF63 and/or pORF63-N+ in the cytoplasm and, to a lesser degree, in the nucleus of VZV-infected pORF63$^{pos}$ keratinocytes in HZ skin biopsies (Fig. 3d). Thus, pVLT-ORF63 and/or pORF63-N+ are expressed in VZV-infected cells in vitro and in vivo.

**Treatment with JNK activator promotes VLT-ORF63 transcription in latently VZV-infected sensory neurons**. We next used our in vitro VZV sensory neuronal latency model[12] to examine transcription patterns across the VLT and ORF63 loci by RT-qPCR during latency and reactivation. Consistent with VZV latency in human TG, VLT was readily detectable in latently VZV-infected HSN, while no other tested viral immediate early (IE), early (E) or late (L) transcripts were detected (Fig. 4a). Notably, we detected neither RNA 63-1 (CDS and 5′-UTR) nor any of the VLT-ORF63 isoforms (Fig. 4a), whose absence was seen only in small fraction of human TGs (Fig. 2a and ref. [7]). 5′-RACE analysis of VLT and ORF63 loci on RNA extracted from latently VZV-infected HSN confirmed expression of core VLT isoform and usage of the two VLT TSS detected in human TG as well as a third TSS located 9 nt upstream of VLT exon 1 (Fig. 2c, orange arrows in bottom row), while $_{lyt}$VLT and $_{lyt}$VLT63 isoforms were not detected.

Similar to previously reported in vitro models for VZV latency using human embryonic stem cell (hESC)-derived neurons[18,19], induction of complete virus reactivation was relatively inefficient in our HSN model[12]. Complete reactivation was observed in 2 of 40 replicates, as demonstrated by the formation of infectious foci after transferring the HSN onto and co-culturing with ARPE-19 cells (Fig. 4b, upper panel), and associated with expression of all IE, E and L VZV transcripts tested in the HSN (Fig. 4c, white

circle). Although no infectious virus was recovered from 38 replicates (Fig. 4b, lower panel), low levels of broad viral gene expression could be detected including VLT63-1 indicative of exit from latency (Fig. 4c, colored triangles). These data indicate that VLT expression is a hallmark of VZV latency in human sensory neurons in vitro, while VLT63-1 and VLT63-2 are induced in response to reactivation stimuli.

Given the critical role of JNK activation as a trigger of HSV-1 lytic gene expression during reactivation[20] as well as the importance of JNK signal in VZV reactivation[21], we speculated that JNK activation may trigger VZV reactivation by inducing lytic gene expression. Latently VZV-infected HSN were treated with the JNK activator anisomycin, a compound that also inhibits protein synthesis. VZV reactivation, as measured by transferring HSN onto ARPE-19 cells for infectious focus forming assay, could not be detected. However, anisomycin treatment consistently increased VLT expression and induced expression of VLT63-1 and VLT63-2 (Fig. 4d). By contrast, anisomycin treatment induced only limited expression of transcripts from the RNA 4 (3 of 6 replicates), RNA 61 (5 of 6 replicates) or RNA 9 (1 of 6 replicates) families, while no lytic VZV gene transcription was detected following mock treatment (Fig. 4e). Thus, anisomycin-mediated JNK activation and/or inhibition of protein synthesis selectively and consistently induces transcription of VLT and both VLT63 isoforms in parallel with or prior to induction of IE gene transcription.

**Ectopic VLT63-1 expression induces broad viral gene expression in the latently VZV-infected sensory neuron model**. The anisomycin-induced VLT63-1 and VLT63-2 transcription suggested that these transcripts, or their encoded proteins, contribute to VZV reactivation. To investigate whether VLT63-1 or VLT63-2 alone can induce VZV reactivation, latently VZV-infected HSN were transduced with replication-incompetent lentivirus vectors encoding ORF63, VLT63-1 or VLT63-2 isoforms, or an empty vector control. Ectopic gene transcription and translation was confirmed in all transduced cell cultures (Fig. 5a and Supplementary Fig. 6). Both in the empty vector and ORF63-transduced cells, endogenous VLT was detected by RT-qPCR at comparable level (Fig. 5a). Sporadic IE and E gene, but no L gene transcription was detected in ORF63- and VLT63-2-transduced cells (Fig. 5b–d). Notably, transcription of VZV IE, E, and L genes was consistently induced by ectopic VLT63-1 expression in latently VZV-infected HSN (Fig. 5b–d). Consistent with the absence or very low level of RNA 62 transcription, which encodes the major viral transactivator protein[22], no infectious VZV was recovered from VLT63-1-transduced HSN cells. Given the differences in protein-coding potential between the VLT63 isoforms and compared to canonical RNA 63-1 (Fig. 3a), the pVLT-ORF63 fusion protein is potentially involved in the transition from latency to reactivation in our HSN VZV latency model.

**Discussion**
VZV latency in human TG is characterized by detection of RNAs expressed from the VLT and occasionally the ORF63 locus, where expression levels of both transcripts correlated positively suggesting co-regulated transcription[7]. Here, we demonstrated that this pattern is best explained by co-expression of VLT and two VLT-ORF63 fusion transcripts, VLT63-1 and VLT63-2, and not by expression of the canonical RNA 63-1 transcripts independent of VLT. Further characterization of the VLT-ORF63 isoforms, expressed during both lytic and latent VZV infection, demonstrated that reactivation stimuli increases VLT and induces VLT63-1 and VLT63-2 expression in our in vitro VZV HSN latency model. Moreover, ectopic VLT63-1 expression induced

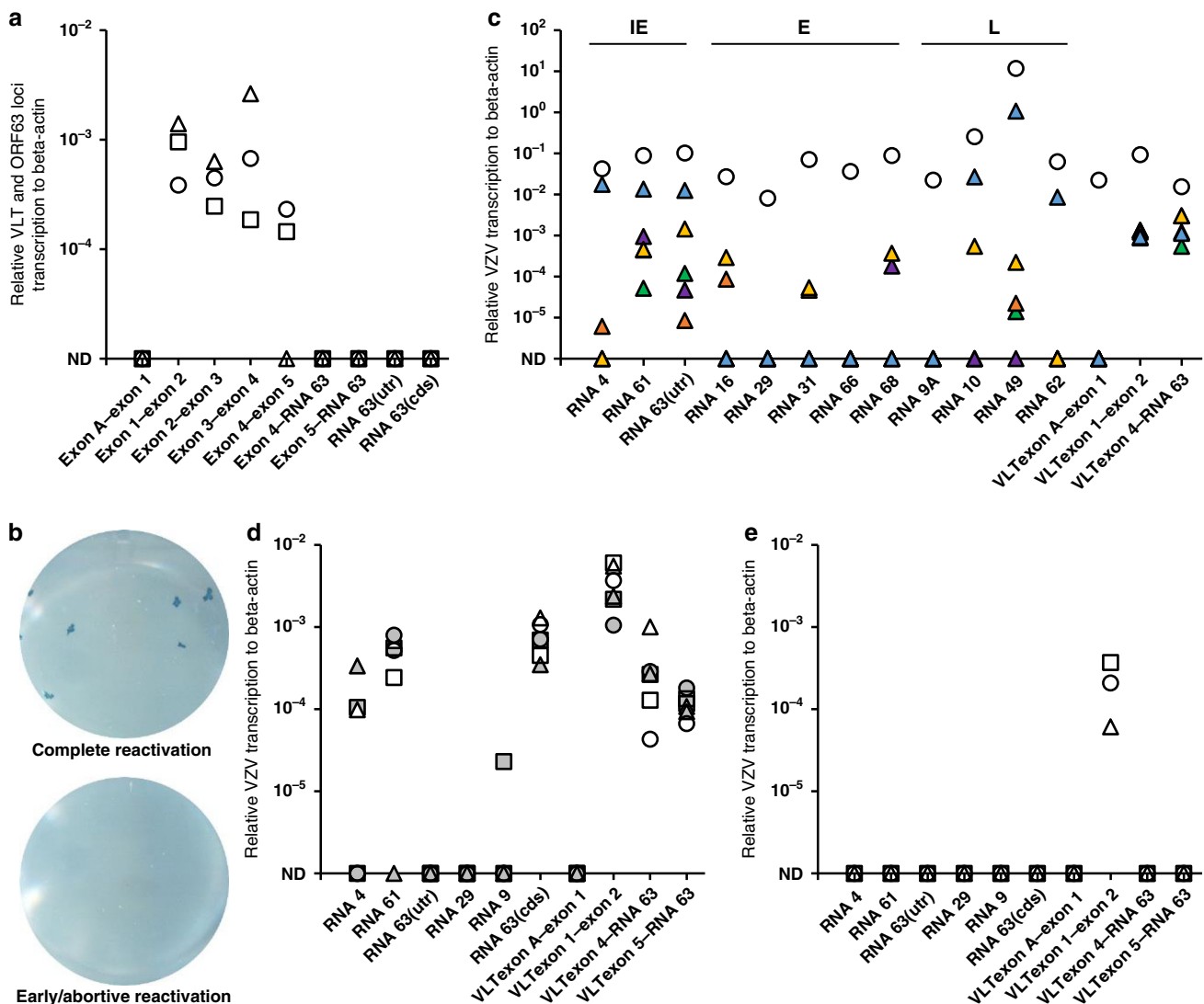

**Fig. 4 Effect of anisomycin treatment on VZV transcription in latently VZV-infected sensory neurons in vitro. a** RT-qPCR analysis for transcription across VLT and ORF63 loci in latently VZV-infected human iPSC-derived sensory neurons (HSN) cultures ($n = 3$). Data on individual HSN culture experiments are shown as unique symbols. **b, c** Latently VZV-infected HSN cultures were depleted of neurotrophic factors (NGF [nerve growth factor], GDNF [glial-derived neurotrophic factor], BDNF [brain derived neurotrophic factor] and NT-3 [neurotrophin-3]) and treated with anti-NGF antibody (Ab) for 14 days. In total, $n = 40$ independent cultures were subjected to reactivation stimuli. **b** Representative examples of a HSN cultures showing complete reactivation (2/40) and early/abortive reactivation (38/40) by infectious focus forming assay after transferring HSN onto ARPE-19 cells. **c** Representative examples of viral gene expression in cultures showing complete reactivation ($n = 1$ shown; white circle) and early/abortive reactivation ($n = 5$ shown; colored triangles). **d, e** HSN cultures were treated with **d** anisomycin ($n = 6$), or **e** DMSO as solvent control ($n = 3$) at both the somal and axonal compartment for 1 h, washed twice and cultured for 7 days before RT-qPCR analysis. Data on individual HSN culture experiments are shown as unique symbols and/or colors. Only VLT exon 1–2 is shown as representative of VLT. Source data are provided as a Source Data file (**a–e**).

broad viral lytic gene transcription in latently VZV-infected HSN, which is potentially mediated by the pVLT-ORF63 fusion protein.

Differential TSS usage of VLT between latently VZV-infected TGs and lytically VZV-infected epithelial cells[7] raises the question as to whether transcription of distinct VLT isoforms depends on infection phase or cell type. Here, we demonstrated that transcription of specific VLT and VLT-ORF63 isoforms is determined by the infection phase rather than the cell type. While both lytic and latent transcript isoforms are predicted to encode the same protein (s), the extended 5′-UTR in the lytic VLT and VLT-ORF63 transcripts may confer new properties to the RNA. Alternative promoter usage of viral genes between distinct but continuous infection phases has been implicated as a switching mechanism of latency and lytic infection among herpesviruses[23–25], and may impact on

the localization and/or function of both type of VZV transcripts at different infection phases.

Transcriptional profiling of reactivating neurons in our in vitro HSN latency model revealed the presence of multiple viral transcripts of all kinetic classes, including the VLT-ORF63 RNAs. By contrast, when attempting to induce reactivation through anisomycin-mediated activation of JNK signaling, previously shown to be required for VZV reactivation and replication[21], only VLT-ORF63 transcription was consistently induced along with higher expression levels of VLT. This provides a favorable explanation as to how VZV initially reactivates from latency. As a counterpoint, we do note the limitations imposed by using anisomycin to show a direct link between JNK signaling and complete reactivation. Specifically anisomycin inhibits protein

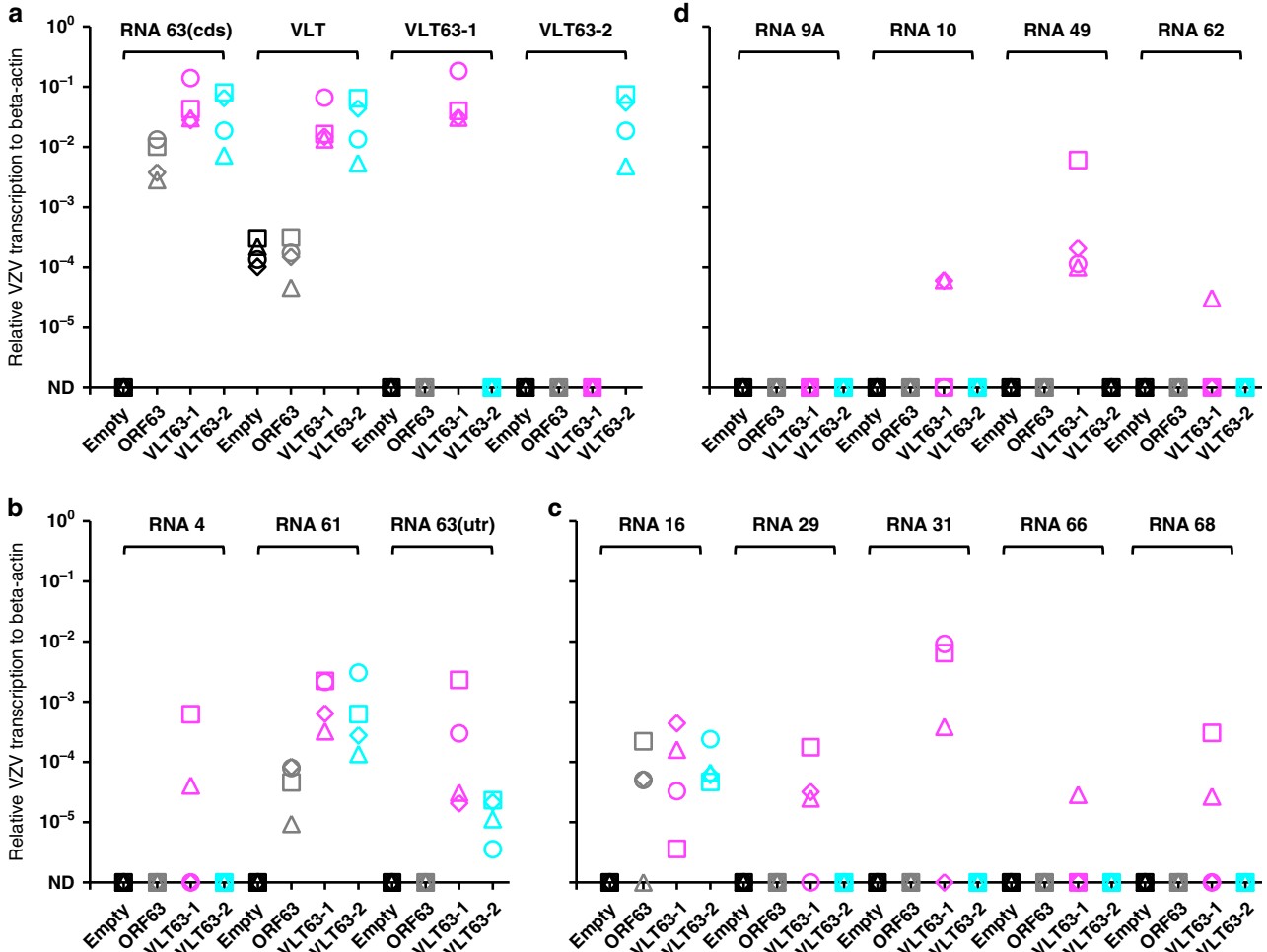

**Fig. 5 Effect of ectopic VLT-ORF63 expression on VZV gene expression in latently VZV-infected sensory neurons in vitro.** At 14 days after establishment of VZV latency in human iPSC-derived sensory neurons (HSN), following VZV genes were transduced by replication incompetent lentivirus vectors: empty vector (white), ORF63 (grey), VLT63-1 (purple) and VLT63-2 (cyan). Transduced HSN were cultured for 14 days (*n* = 4 replicates/vector) and subjected to RT-qPCR analysis. Technical duplicates were utilized per sample and all the biologically independent data is shown as unique symbols for **a** endogenous or transduced VZV genes, **b** IE genes, **c** E genes, and **d** L genes. Source data are provided as a Source Data file.

synthesis and just 1 h exposure to anisomycin induces a terminal decline of neuronal cultures with death occurring within 14 days of treatment, regardless of whether cells were latently VZV-infected or not. Hence, further studies are needed to delineate the mechanisms by which JNK signaling regulates VZV gene expression and reactivation in latently VZV-infected HSN.

The VLT63-1, but not VLT63-2 isoform induced broad lytic viral gene transcription in latently VZV-infected HSN cultures. While these two isoforms differ only by the partial presence/absence of VLT core exon 5, only VLT63-1 encodes for the pVLT-ORF63, suggesting that the encoded protein and not the transcript itself regulates transcription of VZV genes. Canonical pORF63 has been reported to both positively and negatively affect VZV gene transcription during lytic infection[26,27], depending on its phosphorylation status and cell type analyzed[28]. Here, we showed that pORF63 induces expression of IE transcript RNA 61-1 encoding the promiscuous transactivator pORF61[29] and the E transcript RNA 16-1, but not any of the other analyzed VZV IE, E, or L genes in our in vitro HSN latency model. By contrast, pVLT-ORF63 induced expression of multiple VZV transcripts of all kinetic classes. Notably, in addition to its transcriptional regulatory activities, VZV pORF63 was reported to protect neurons from apoptotic cell death and to antagonize innate antiviral immune responses[30–32]. Further studies are required to

determine whether pVLT-ORF63 similarly modifies apoptosis and immune pathways. Overall, the data suggest a key role for pVLT-ORF63 in the initiation of VZV reactivation and emphasize the importance of uncharacterized cellular factors and/or other viral factors that are required for complete VZV reactivation for future studies.

Endogenous pVLT-ORF63 protein expression in human TG and in vitro HSN latency model was analyzed, however, IHC staining using the polyclonal chicken anti-pVLT-ORF63 antibody as well as the polyclonal rabbit anti-pVLT antibody was unfortunately found to result in high levels of non-specific background staining in neuronal tissue (staining with the polyclonal rabbit anti-pVLT antibody in HSN is shown in Supplementary Fig. 6). Notably, in the past we have stained substantial numbers of human TG using a highly potent pORF63-specific mAb (which will recognize both cognate pORF63 and pVLT-ORF63) and never observed any specific pORF63/pVLT-ORF63 staining[33], while others reported that detection of pORF63/pVLT-ORF63 is extremely rare and possibly associated with reactivation[34]. Combined, the data indicate that either pVLT-ORF63 is not expressed or expressed at very low quantities, below the sensitivity of IHC, in latently VZV-infected human TG and HSN.

HSV reactivation has been proposed to consist of two distinct but continuous waves of lytic gene transcription. The first wave

designated as animation[35] or Phase I[36,37] is characterized by generalized viral gene derepression[36,38], in contrast the second wave (Phase II) is identical to the cascade observed during acute infection. Transition from latency to Phase I and Phase I to Phase II might work as checkpoints, determining whether the virus should complete reactivation or halt reactivation and re-enter latency[36]. While we do not know whether animation/Phase I occurs during VZV reactivation, broad transcription of viral genes of all kinetic classes was induced by reactivation stimuli, occasionally leading to complete reactivation in our in vitro HSN latency model. Notably, our data demonstrates that broad VZV transcription is initiated by VLT-ORF63 transcription/pVLT-ORF63 translation, unlike Phase I of HSV reactivation for which no viral initiator has been proposed to drive HSV gene expression. Thus, there might be critical differences between the mechanism(s) governing HSV and VZV latency and reactivation.

Taken together, our discovery of the VLT-ORF63 fusion transcripts in VZV-infected human TG and the ability of the encoded pVLT-ORF63 fusion protein to act as an initiator of viral gene expression during reactivation in our in vitro HSN latency model provides critical new insights into how VZV latency and reactivation are governed. Further studies using the HSN VZV latency model, combined with careful analyses of human ganglia, provides a new platform to dissect the cellular and viral molecular mechanisms controlling VZV latency and reactivation. VZV is the only human herpesvirus for which vaccines are licensed. However, the live-attenuated varicella vaccine establishes latency and may reactivate to cause disease, while adjuvanted recombinant zoster vaccine is highly effective against HZ prevention but not suited for childhood varicella vaccination. The discovery of VLT-ORF63 and its role in reactivation provides new insight that will inform efforts to improve varicella vaccines.

## Methods

**Human clinical specimens**. Human TG specimens were obtained (Supplementary Tables 1 and 2) from the Netherlands Brain Bank (Netherlands Institute for Neuroscience; Amsterdam, the Netherlands). All donors had provided written informed consent for brain autopsy and the use of material and clinical information for research purposes. All study procedures were performed in compliance with relevant laws in The Netherlands and Japan, institutional guidelines approved by the local ethical committee (VU University Medical Center, Amsterdam, project number 2009/148, and Kobe University, Kobe, project number 170107) and in accordance with the ethical standards of the Declaration of Helsinki. TG biopsies were either formalin-fixed and paraffin-embedded (FFPE) for in situ analysis or snap-frozen in liquid nitrogen and stored at −80 °C for nucleic acid extraction. FFPE skin punch biopsies of one healthy control subject and two herpes zoster skin lesions were obtained for diagnostic purposes. According to the institutional "Opt-Out" system (Erasmus MC, Rotterdam, the Netherlands), which is defined by the National "Code of Good Conduct" [Dutch: Code Goed Gebruik, May 2011], the surplus human herpes zoster FFPE tissues were available for the current study.

**Cells**. Human iPSC-derived sensory neuron (HSN) progenitors (Axol Bioscience) were plated on a 24-well plate ($1 \times 10^5$ cells/well), CELLview Slide (Greiner Bio-One) ($1 \times 10^4$ cells/well) or a microfluidic platform ($7.5 \times 10^4$ cells/sector) in Neuronal Plating-XF Medium (Axol Bioscience). Fabrication of a microfluidic platform was previously described[39]. Prior to plating the HSN progenitors, a plate, slide or microfluidic platform was coated with poly-L-ornithine (Sigma-Aldrich) (20 μg/mL) or poly-D-lysin (Sigma-Aldrich) (200 μg/mL) in molecular grade water at room temperature overnight, washed with distilled water twice and coated with Matrigel (Corning) (1 μg/mL) in Knockout DMEM/F-12 medium (Thermo Fisher Scientific) for 2 h at room temperature following overnight incubation at 37 °C in a humified 5% $CO_2$ incubator. At 1 day after plating, the medium was replaced to the complete maintenance medium consisted with Neurobasal Plus Medium, B-27 Plus Supplement (2% [vol/vol]), N2 Supplement (1% [vol/vol]), GlutaMAX-I (2 mM) (Thermo Fisher Scientific), ascorbic acid (200 μM; Sigma-Aldrich), GDNF (glial-derived neurotrophic factor) (25 ng/mL), NGF (nerve growth factor) (25 ng/mL), BDNF (brain derived neurotrophic factor) (10 ng/mL) and NT-3 (neurotrophin-3) (10 ng/mL) (Peprotech) for sensory neuronal maturation. Two days after the plating the HSN progenitors, cells were treated with the complete maintenance medium with mitomycin C (2.5 μg/mL; Nacalai Tesque, Inc.) for 2 h to eliminate proliferating cells, washed with the complete medium twice and culture in the complete maintenance medium at least 7 weeks with replacing half the volume of

culture with the fresh medium every 4 days. During maturation in the microfluidic platform, culture medium level in the axonal compartment was kept higher than that in the somal compartment to prevent cell migration to the axonal compartment. The maturation of sensory neurons was characterized previously[12]. Human retinal pigmented epithelium ARPE-19 cells (American Type Culture Collection [ATCC] CRL-2302) were maintained in DMEM/F-12+GlutaMAX-I (Thermo Fisher Scientific) supplemented with heat-inactivated 8% FBS (foetal bovine serum; Sigma-Aldrich). Human embryonic kidney (HEK) 293T cells (ATCC CRL-3216) were cultured in DMEM + GlutaMAX-I (Thermo Fisher Scientific) supplemented with heat-inactivated 8% FBS.

**VZV infections**. VZV strain pOka (parental Oka) was maintained in and the cell-free virus was prepared from ARPE-19 cells as described previously for human embryonic lung fibroblast MRC-5 cells[40]. For lytic infection in ARPE-19 cells, cells were plated on 12-well plate at a density of $1 \times 10^5$ cells/well 2 days before infection, infected with the cell-free virus ($3 \times 10^3$ pfu [plaque-forming unit] to 1 well) for 1 h in 500 μL medium, washed with the medium twice and cultured for 4 days. For lytic infection for HSN, cells were matured as described above, infected with the cell-free virus ($4 \times 10^3$ pfu to 1 well) at 52 days after maturation for 2 h in 400 μL medium, washed with the medium twice, treated with low pH buffer (40 mM sodium citrate, 10 mM potassium chloride, 135 mM sodium chloride [pH 3.2]) for 30 seconds (sec), washed with the medium once and cultured for 2 weeks to obtain efficient lytic infection as described[12].

For in vitro VZV latency, the method was established previously for human embryonic stem cell-derived neurons[19] and applied for HSN[12]. Briefly, HSN were matured on a microfluidic platform for 54 days and infected via axonal chamber with 10 μL of the cell-free virus ($4 \times 10^4$ pfu/mL titrated on ARPE-19 cells) in 20 μL total volume. After 2 h infection, inoculum was removed, and axons were treated with the low pH buffer for 30 sec, washed with the medium and cultured for 2 weeks. To reactivate VZV from sensory neuronal latency, GDNF, NGF, BDNF, and NT-3 were depleted from and anti-NGF polyclonal antibody (50 μg/mL) was added to the complete maintenance medium, and cultures were maintained for 2 weeks. For chemical induced reactivation, latently VZV-infected HSN was treated with JNK activator, anisomycin (20 μg/mL) or solvent control (DMSO [dimethyl sulfoxide]) for 1 h from both axonal and somal compartment, washed with the medium twice and cultured in the complete maintenance medium without GDNF, NGF, BDNF and NT-3 for 1 week. For VZV gene transduction, latently VZV-infected HSN was transduced by lentivirus vector for 2 h from somal compartment with mixing by pipetting at 30 minutes (min) and 1.5 h after transduction, cultured overnight and replaced the medium in both somal and axonal compartments with the fresh complete maintenance medium without GDNF, NGF, BDNF and NT-3 for 2 weeks. The complete reactivation was confirmed by the formation of infectious foci after transferring the HSN onto and co-culturing with ARPE-19 cells for 7 days as described below.

**DNA, RNA, and cDNA**. DNA and RNA were isolated from human TGs ($n = 4$) (Supplementary Table 2) as described previously[7]. DNA and RNA from VZV-infected ARPE-19 cells or HSN were isolated as described previously[7] with slight modifications using the FavorPrep Blood/Cultured Cell Total RNA Mini Kit (FAVORGEN BIOTECH) in combination with the NucleoSpin RNA/DNA buffer set (Macherey-Nagel). DNA was first eluted from the column in 80 μL DNA elution buffer, the column was treated with recombinant DNase I (20 units/100 μL; Roche Diagnostics) for 30 min at 37 °C and RNA was eluted in 50 μL nuclease free water. RNA was directly treated with Baseline-ZERO DNase (2.5 units/50 μL; Epicentre) for 30 min at 37 °C (all the RNA), and further purified by Dynabeads mRNA purification kit (Thermo Fisher Scientific) (TG RNA) or enriched by Agencourt RNAClean XP (Beckman Coulter) (in vitro latency RNA). cDNA was synthesized with 12 μL of RNA and anchored oligo(dT)$_{18}$ primer in a 20 μL reaction using the Transcriptor First Strand cDNA synthesis kit at 55 °C for 30 min for reverse transcriptase reaction (Roche Diagnostics).

**Quantitative PCR and 5′-RACE analysis**. DNA or cDNAs were subjected to quantitative PCR (qPCR) using KOD SYBR qPCR Mix (TOYOBO) in the StepO-nePlus Real-time PCR system (Thermo Fisher Scientific) (1 μL of DNA or cDNA per 10 μL reaction in duplicate). All the primer sets used for qPCR (Supplementary Table 1) were first confirmed for the amplification rate (98-100%) and their own melting temperatures using $10-10^6$ copies (tenfold dilution) of pOka-BAC genome, VLT-ORF63 plasmids or purified PCR product spanning from VLT exon A to exon 1 of cDNA and the lack of non-specific amplification using water (except a primer pair for RNA 31 which signal was detected with Ct value of 36.25 cycles with distinct melting temperature [69.79 °C] from authentic ORF31 amplicon [80.97 °C]). Due to the partially antisense nature of VLT via exon 3 and exon 4 against RNA 61, a primer pair of exons 3 and 4 detected both VLT and RNA 61, thus a primer pair of exons 2 and 4 was used for lytic infection instead a primer pair of exons 3 and 4 used for in vivo and in vitro latency in which RNA 61 is absent. The qPCR program is as follows; 95 °C for 2 min (1 cycle), 95 °C for 10 sec and 60 °C 15 sec (40 cycles), and 60 to 95 °C for a dissociation curve analysis to discriminate non-specific signal if any. Data are presented as relative transcription level of VZV gene to cellular beta-actin defined as $2^{-(Ct\text{-}value\ VZV\ gene\ -\ Ct\text{-}value\ beta\text{-}actin)}$.

5′-RACE analysis was performed using SMARTer RACE 5′/3′ kit and In-Fusion HD Cloning kit according to the manufacturer's instructions with slight modifications (Clontech). 5′-RACE ready cDNA was synthesized from purified mRNA (TG) or total RNA (HSN) with SMARTerIIA Oligonucleotide and 5′-RACE CDS Primer A. KOD FX Neo PCR system (TOYOBO) was used for 5′-RACE PCR and the program was 1 cycle of 94 °C for 2 min and 30 cycles of 95 °C for 10 sec and 68 °C for 1 min. Initial PCR was performed by Universal Primer Mix (a mixture of Universal Primer Long and Universal Primer Short) and InFusionVLTexon5R104799 for VLT or InFusionORF63R805 for VLT-ORF63 fusions. Nested PCR, if necessary, was carried out using Universal Primer Long and InFusionVLTexon4R10361 for VLT or InFusionORF63R622 for VLT-ORF63 fusions. The 5′-RACE PCR products were cloned into linearized pRACE cloning vector and sequenced by M13forward or M13reverse on the ABI Prism 3130 XL Genetic Analyzer using the BigDye v3.1 Cycle Sequencing Kit (Thermo Fisher Scientific). All the primers used for 5′-RACE analysis are listed in Supplementary Table 4.

**Direct RNA sequencing and cDNA sequencing on nanopore array**. Direct RNA sequencing libraries were generated from 117 to 153 ng of poly(A) RNA, isolated using Dynabeads mRNA purification kit. Isolated poly(A) RNA was subsequently spiked with 0.5 μl of a synthetic Enolase 2 (ENO2) calibration RNA (Oxford Nanopore Technologies Ltd.) and sequenced on a MinION MkIb with R9 flow cells (Oxford Nanopore Technologies Ltd.) for 40 h, as previously described[14]. cDNA sequencing libraries were generated from 1 ng of poly(A) RNA, isolated using Dynabeads mRNA purification kit using cDNA-PCR Sequencing kit (SQK-PCS109) (Oxford Nanopore Technologies Ltd.) and sequenced on a MinION MkIb with R9 flow cells for 40 h using the command software MinKNOW v3.1.9 (Oxford Nanopore Technologies). Following basecalling with Guppy v3.2.2, only the reads passing filter were used. Error-correction was performed using proovread as described previously[14]. Nanopore read data were aligned to the VZV strain dumas genome (X04370.1) using MiniMap2 v2.15[41] and parsed using SAMtools v1.9[42] and BEDTools v2.27.1[43] before visualizing using the Bioconductor v3.11 and Gviz v1.32.

**Multiplex fluorescent RNA in situ hybridization in human cadaveric TG**. FFPE latently VZV-infected human TG (n = 7) (Supplementary Table 3), human zoster skin biopsies, and lytically VZV-infected ARPE-19 cells were analyzed by multiplex fluorescent in situ hybridization (mFISH) using the RNAScope Multiplex Fluorescent Reagent Kit v2 (Advanced Cell Diagnostics) according to the manufacturer's instructions. Briefly, deparaffinized 5 μm-thick tissue sections were incubated with probes directed to VZV RNA 9, RNA 63 and VLT exons 2–5. RNA integrity and mFISH specificity were demonstrated by staining TGs for ubiquitously expressed cytoplasmic transcript UBC (human ubiquitin C), nuclear transcript MALAT1 (human metastasis-associated lung adenocarcinoma transcript 1) (positive controls) and bacterial dihydrodipicolinate reductase (DAPB) (negative control probe) (Supplementary Fig. 2C). Probes were designed by Advanced Cell Diagnostics. Sections were mounted with Prolong Diamond antifade mounting medium and confocal microscopic analysis was performed on a Zeiss LSM 710 confocal microscope, as described[44].

**Replication incompetent lentivirus vectors**. VZV genes, ORF63, VLT63-1, or VLT63-2 were amplified by PCR of cDNA from VZV pOka-infected ARPE-19 cells. The PCR product was digested with SalI restriction enzyme and cloned into CS-CA-MCS plasmid (Riken BioResource Research Center) via SalI site (ORF63) or directly cloned into linearized CS-CA-MCS (VLT63-1 and VLT63-2) using In-Fusion HD Cloning kit according to the manufacturer's instruction (Clontech). The primer sets used for cDNA cloning into the CS-CA-MCS plasmid are listed in Supplementary Table 4. HEK293T cells (4 × 10⁶ cells) were plated onto a 10 cm dish, transfected with CS-CA-MCS or CS-CA-VZV (20 μg) and packaging plasmids, pCAG-HIVgp (5 μg) and pCMV-VSV-G-RSV-Rev (5 μg) (Riken BioResource Research Center) using PEImax solution (60 μL) (Polysciences) prepared in KnockoutDMEM/F-12 (500 μL) at 6 h post plating and cultured overnight in 8% $CO_2$ incubator in 10 mL DMEM + GlutaMAX-I with heat-inactivated 8% FBS. Whole culture medium was changed with the fresh medium at 16 h after transfection and cells were cultured for another 48 h. Supernatant (20 mL) were filtrated through 0.45 μm syringe filter (Pall), mixed with 5 mL of PEG6000 (50% [wt/vol] in PBS), 1.7 mL of 5 M NaCl and 2.6 mL of PBS, and rotated in a 50 mL conical tube at 4 °C at least 8 h. After centrifugation at 7000 × g for 10 min, supernatant was removed, pellet was resuspended in 300 μL of KnockoutDMEM/F-12, aliquoted out in four tubes and stored at −80 °C until use. Quantity of each replication incompetent lentivirus vector was measured by qPCR of cDNA synthesized with random hexamer from genomic RNA packaged in enriched pseudo virion using the primer set, CSCA1831F and CSCA1969R (Supplementary Table 1) detecting upstream promoter region in CS-CA-MCS plasmid and equal amount of virus were used for transductions.

**Antibodies**. Chicken polyclonal antibody (pAb) against 24 aa linker peptide (GFVRFITRQRRVGFKGKGYYGPKD) of pVLT-ORF63 fusion protein, anti-pVLT-ORF63 pAb was generated and purified through an immunogen conjugated

peptide column (Cosmo Bio). Rabbit anti-pVLT pAb, rabbit anti-pORF63 pAb[7], mouse anti-pORF63 monoclonal antibody (mAb)(clone VZ63.08)[45], and mouse anti-glycoprotein E (gE) mAb[46] were described previously. Anti-alpha tubulin mAb (clone B-5-1–2; Sigma-Aldrich) and sheep anti-NGF pAb (EMD Millipore) are commercially available. Alexa Fluor 488- or Alexa Fluor 647-conjugated donkey anti-mouse IgG, Alexa Fluor 594-conjugated donkey anti-rabbit IgG (Thermo Fisher Scientific) and Alexa Fluor 488-conjugated donkey anti-chicken IgY (Jackson ImmunoResearch Laboratories) were used for secondary Abs for indirect immunofluorescent assay. Anti-mouse IgG HRP-linked Whole Ab Sheep or anti-rabbit IgG HRP-linked Whole Ab Donkey (GE Healthcare Bio-Sciences) were used as secondary Abs for immunoblotting.

**Immunofluorescent staining, confocal microscopy, infectious foci staining, and immunoblotting**. Cells on CELLview slide were fixed with 4% (vol/vol) paraformaldehyde (PFA)/PBS (Nacalai Tesque, Inc.) at room temperature for 20 min, permeabilized with 0.1% Triton X-100/4% PFA/PBS at room temperature for 20 min, and incubated with human Fc receptor blocking solution (5% FBS/PBS containing 10% of Clear Back [MBL]) at room temperature for 1 h. Cells were stained with the primary Abs diluted in a solution (5% FBS/PBS) overnight at 4 °C (1:100 for anti-pVLT pAb, anti-pVLT-ORF63 pAb and anti-pORF63 mAb, 1:300 for all Abs against neuronal markers, 1:500 for anti-pORF63 pAb), washed with 0.1% Tween 20/PBS (PBS-T) for 5 min 3 times, stained with the secondary Abs (1:300) diluted in 5% FBS/PBS at room temperature for 1 h, washed with PBS-T for 5 min 3 times, covered with VECTASHIELD Vibrance Antifade Mounting Medium with DAPI (Vector Laboratories) and imaged by an FV1000D confocal microscopy (Olympus).

Deparaffinized and rehydrated 5 μm FFPE sections of human herpes zoster skin lesions and healthy control skin were subjected to heat-induced antigen retrieval with citrate buffer (pH = 6.0), blocked and incubated with mouse anti-VZV pORF63 Ab (1:1,500 dilution; kindly provided by Dr. Sadzot-Delvaux; Liege, Belgium)[47]), chicken anti-pVLT-ORF63 pAb (1:100 dilution) overnight at 4 °C. Sections were subsequently incubated with Alexa Fluor 488- and Alexa Fluor 594-conjugated goat-anti-mouse and goat-anti-chicken secondary antibodies (all 1:250 dilution) and sections were mounted with Prolong Diamond antifade mounting medium with DAPI. Confocal microscopic analysis was performed as described[44].

To visualize infectious foci on ARPE-19 cells, cells were fixed with 4% PFA/PBS, stained with anti-gE mAb (1:10 dilution in PBS), followed by anti-mouse IgG HRP-linked whole Ab sheep (1:5,000 dilution in PBS), and reacted with 3, 3′, 5, 5′-tetramethylbenzidine-H peroxidase substrate (Moss, Inc.).

Cells were incubated in RIPA lysis buffer (0.01 M Tris-HCl [pH 7.4], 0.15 M NaCl, 1% sodium deoxycholate, 1% Nonidet P-40 and 0.1% SDS) on ice for 15 min, sonicated in a water bath for 10 min, centrifuged at 20,000 × g for 15 min. Supernatant was boiled with LDS Sample Buffer (4X) and Sample Reducing Agent (10X) at 100 °C for 5 min (Thermo Fisher Scientific). Proteins were separated on 4-12% Bis-Tris Plus Gel in MES SDS Running Buffer (200 V, 25 min), transferred onto PVDF membrane (0.2 μm) using Mini Blot Module (20 V, 1 h) in Bolt Transfer Buffer containing 10% methanol and 0.1% Bolt Antioxidant (Thermo Fisher Scientific). The membrane was blocked in a blocking buffer (5% [wt/vol] skimmed milk/0.1% Tween 20/PBS) at room temperature for 1 h, stained with primary Abs diluted in the blocking buffer (1:1,000 for anti-pVLT pAb, 1:6,000 for anti-pORF63 pAb and 1:10,000 for anti-alpha tubulin mAb) overnight at 4 °C, washed with PBS-T for 5 min 3 times, stained with the secondary Abs diluted in the blocking buffer (1:3,000) at room temperature for 30 min, and washed with PBS-T for 5 min 3 times and PBS briefly once. Signals were visualized by Chemi-Lumi One Super (Nacalai Tesque, Inc.) and captured using LAS4000mini (GE Healthcare Bio-Sciences). A membrane stained with anti-pVLT pAb was stripped by WB Stripping Solution Strong in accordance with the manufacturer's manual (Nacalai Tesque, Inc.) and reprobed with anti-alpha-tubulin mAb.

**Reporting summary**. Further information on research design is available in the Nature Research Reporting Summary linked to this article.

## Data availability
Basecalled fast5 nanopore dRNA- and cDNA-Seq datasets generated as part of this study can be downloaded from the European Nucleotide Archive (ENA) under the following study accession: PRJEB36978. Source data are provided with this paper.

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

## Acknowledgements

We are grateful to Yasuko Mori for use of laboratory equipment, Angus C. Wilson and Ian Mohr for critical reading of the manuscript. We thank Tamana Mehraban for technical assistance. This work was supported by the Takeda Science Foundation, Daiichi Sankyo Foundation of Life Science, Japan Society for the Promotion of Science (JSPS KAKENHI JP17K008858, JP16H06429, and JP16K21723) and the Ministry of Education, Culture, Sports, Science and Technology (MEXT KAKENHI JP17H05816) (T. S.). Research reported in this publication was also supported by the National Institute of Allergy and Infectious Diseases of the National Institutes of Health under Award Number R01AI151290 (W.J.D.O. and G.M.G.M.V.). The content is solely the responsibility of the authors and does not necessarily represent the official views of the National Institutes of Health. D.P.D. was supported in part by NIH grants (R01GM056927, R01AI073898, R01AI152543). A.V. is supported by the NIH NINDS (R21NS107991). J.B. receives funding from the UCL/UCLH NIHR Biomedical Research Centre. S.J. and T.L.R. are supported by "Strengthening the capacity of CerVirVac for research in virus immunology and vaccinology", grant no. KK.01.1.1.01.0006, awarded to the Scientific Centre of Excellence for Virus Immunology and Vaccines and co-financed by the European Regional Development Fund. The funders had no role in study design, data collection, and interpretation, or in the decision to submit the work for publication.

## Author contributions

W.J.D.O., D.P.D., and T.S. designed the study, performed the experiments and analyzed the data. W.J.D.O., D.P.D., G.M.G.M.V., and T.S. wrote the paper. L.R., T.L.R., S.J., and A.V. contributed to new reagents/analytic tools. J.B. and G.M.G.M.V. provided additional input into the study. All authors read and approved the final paper.

## Competing interests

The authors declare no competing interests.
