## [Peer Review File · Nature Communications]

Reviewer comments, first round -

Reviewer #1 (Remarks to the Author):

This manuscript describes the discovery and characterization of varicella-zoster virus transcripts produced in latently infected cells. The work is a marvelous expansion of their earlier identification of VZV latency transcript (VLT) which VZV was previously believed to lack (in contrast to other alphaherpesviruses). Both VLT and ORF63 transcripts had previously been described in latently infected neurons, but the current work determined that ORF63 transcript is present in those cells only as VLT-ORF63 splice transcripts. One of several splice transcripts, VLT-ORF63a, encodes a unique protein; they also provide evidence in an in vitro neuronal culture system that this protein likely plays a direct role in the initiation of VZV gene transcription, thus driving the virus into a lytic state. This represents a very significant step forward in our understanding of how latency and reactivation are regulated by VZV. The work performed was meticulous and impressive. The manuscript is lucidly and concisely written. I have no recommendations for improving the manuscript.

Reviewer #2 (Remarks to the Author):

This study by Ouwendjik & Depledge et al demonstrates the existence of a novel fusion transcript in VZV (VLT-ORF63) that appears to be important for reactivation of the virus from latency. They use a number of complementary molecular biology techniques (dRNA-seq, cDNA seq, RT-qPCR and RACE), to clearly demonstrate and map the novel transcript during lytic replication and latent infection of a novel model system in addition to human TGs. Employing both RT-qPCR and dual RNA-FISH they demonstrate the existence of the transcript in natural latency (TGs isolated from human autopsy material). Interesting, the transcript appears to be absent from the in vitro model of latency. The authors produce antibodies that specially recognize the resultant protein to demonstrate its synthesis during lytic replication, but disappointingly do not use this antibody to probe latent samples nor following reactivation. Finally, they show that reactivation of VZV from in vitro infected differentiated human sensory neurons results in VLT-ORF63 transcription. In addition, transduction with VLT-ORF63 promotes reactivation from latency. In sum, the combined systems and approaches used generate a convincing study showing the presence of this novel transcript in VZV latency and/or reactivation. However, its role in reactivation could be strengthened, especially considering that these experiments do rely on a small number of experimental replicates.

Major points.

1. A number of experiments seem to be underpowered and therefore lack rigor. There is no mention as to how power analysis was carried out. In particular for figures 4 and 5 using the differentiated human neurons, only 2 or 3 biological replicates have been carried out.
2. Can the authors use their antibodies to investigate VLT-ORF63 protein synthesis in the human TGs or during reactivation in the in vitro model system? The localization of the mRNA in the human TGs is interesting given its predominantly nuclear localization. The authors do not comment on this. Does this suggest that protein is not translated during latency?
3. The in vitro infected neurons undergo weak reactivation using stimuli such as neurotrophic factor deprivation, anisomycin or VTL-ORF63 transduction. Have the authors investigated whether a combination of anisomycin and VLT-ORF63 enhances reactivation?

Minor points

1. Can the authors include the limit of detection information for the RT-qPCR primers used to be sure this is not differential expression of different transcripts in latency is not a detection issue?
2. More quantification of the FISH experiment would good showing the numbers of positive neurons for each probe, the numbers colocalizing verses present in the same neuron but distinct.
3. Lns 199-211. This paragraph is confusing in the way that it is written. First "the same result" in Ln 202 is unclear given that two results are mentioned in the preceding sentence. Can they also comment more on the differential localization – it looks like following transfection ORF63 and VLT

are mainly cytoplasmic proteins, whereas the VLT-ORF63 appears to be nuclear. However, during lytic it is nuclear and cytoplasmic.

4. In figure 4c it's not entirely clear what the black and white circles represent. Is this just one N for each or the means from the combined 40 samples? The text combined with the legend is confusing. The text mentions that these are the samples that did not reactivate but the legend calls them early reactivation events. Perhaps these are abortive reactivations instead of early?

5. Can the authors do a direct comparison between IE, E and L transcripts in the differently transduced samples? Does VLT-ORF63 specifically promote the viral DNA replication and/or transition to late gene expression?

Reviewer #3 (Remarks to the Author):

This is an important paper that reveals the existence of novel fusion transcripts, which combine the VZV latency-associated transcript (VLT) with the virus immediate early ORF63 gene. The fusion transcript (VLT-ORF63) is expressed during both lytic and latent infection and is shown to be expressed in human trigeminal ganglia. The authors demonstrate that one of the two major VLT-ORF63 isoforms encodes a novel fusion protein (pVLT-ORF63) and the functional significance of this novel protein is investigated in vitro using iPSC derived human sensory neuronal cell cultures. Using this system it is shown that whilst VLT is transcribed during latency, VLT-ORF63 can be induced by the application of reactivation stimuli. Subsequent studies reveal that expression of pVLT-ORF63 induced transcription of VZV lytic gene expression. Although full reactivation resulting in the production of infectious virus progeny was not observed in this system, the ability of pVLT-ORF63 to reverse transcriptional silencing of the latent VZV genome supports the authors view that expression of a novel VLT-ORF63 fusion transcript is likely to play a role in the transition from latency to lytic infection.

I have only a few comments.

1. The detection of VLT-ORF63 transcripts in human trigeminal ganglia adds to the significance of paper but further explanation and consideration of the results are required. In Figure 2a it is not clear why there is a negative RT-qPCR signal for VL Texon A-1 primer set. This would rule out detection of Lytic VLT, Lytic VLT-ORF63a and Lytic VLT-ORF63b transcripts (supplementary Figure 1). This left me confused. Which VLT-ORF63 fusion transcripts are detected in human ganglia? I appreciate that the situation is clarified in part by the inclusion of 5'RACE data but it would be helpful if the characteristics of the lytic and latent VLT transcripts were discussed earlier in the results section.

2. Figure 2b. The in situ hybridisation data on sections of human trigeminal ganglia confirmed expression of VLT or co-expressed nuclear VLT and ORF63 RNA. It would be helpful if the authors could clarify the proportion of neurons expressing VLT alone and give some indication whether VLT is expressed in all latent DNA containing cells. Is VLT a solid marker for latently infected cells? If the functional data presented favours the view that VLT-ORF63 fusion transcripts encode a function necessary for virus reactivation, is the detection of neurons expressing co-localizing ORF63/VLT in situ signal revealing cells at early stages of reactivation? Do these neurones express lytic gene transcripts?

3. The VZV sensory neuronal cell culture system provides a useful model to study VZV latency and reactivation and a key feature of this system is the establishment of a form of tight latency in which VLT is the sole transcript expressed. This appears to differ from the results presented for human trigeminal ganglia where the expression of both VLT and VLT-ORF63 fusion transcripts are commonly detected (compare Figure 2a with Figure 4a). Nonetheless, application of a reactivation stimulus by depletion of neurotropic factors resulted in the detection of multiple viral transcripts including VLT-ORF63a. Anomycin treatment of cultures also induced expression of VLT and VLT-ORF63 isoforms but with more limited induction of lytic gene transcription. In order to determine whether induced transcription of VLT-ORF63 transcripts is directly involved in genome de-repression latently infected cultures were transduced with Lentivirus vectors expressing either ORF63 or the latent VLT-ORF63a or b isoforms. Both ORF63 and VLT-ORF63b transduced cells resulted in sporadic IE and early gene activation whilst transduction with VLT-ORF63a resulted in the activation of all gene classes. In no instance was full virus reactivation observed. This result

has tempered the conclusions, which proposes a potential role for the pVLT-ORF63 fusion protein in the transition from latency to reactivation. Given that the VLT-ORF63 isoforms are detected during latency in human trigeminal neurones, it may be premature at this stage rule out other potential functions of novel VLT-ORF63 isoforms that may relate to the known anti-apoptotic functions of ORF63 (Hood et al 2005., Gerada et al., 2018). Such a neuronal survival mechanism may be important both for stable maintenance of VZV latency and to facilitate efficient reactivation.

We would like to thank the reviewers and editor for their enthusiasm toward our work, and for their comments and critiques which have provided us with a platform to further strengthen our manuscript. Below we offer detailed point-by-point responses to each comment/critique.

Besides the points raised by the reviewers, during preparation of this revised manuscript, we published our manuscript describing the reannotation of the lytic VZV transcriptome and kinetic classes of VZV transcripts (accepted in mBio; reference 15 for bioRxiv version). For consistency we have therefore updated the nomenclature of canonical VZV transcripts and redefined the lytic isoforms of VLT, VLT-ORF63a, VLT-ORF63b, VLT-ORF63c as *lyt*VLT, *lyt*VLT63-1, *lyt*VLT63-2, *lyt*VLT63-3, respectively. The corresponding transcript isoforms expressed during latency are defined as VLT, VLT63-1 and VLT63-2 (*lyt*VLT63-3 does not have a latent isoform). While our reannotation redefined the kinetic class of some VZV genes used in our (original and revised) VZV VLT-ORF63 manuscript, these changes did not affect the overall results in our current manuscript. Please note that all changes to the text in the manuscript are highlighted in red.

Reviewer #1 (Remarks to the Author):

This manuscript describes the discovery and characterization of varicella-zoster virus transcripts produced in latently infected cells. The work is a marvelous expansion of their earlier identification of VZV latency transcript (VLT) which VZV was previously believed to lack (in contrast to other alphaherpesviruses). Both VLT and ORF63 transcripts had previously been described in latently infected neurons, but the current work determined that ORF63 transcript is present in those cells only as VLT-ORF63 splice transcripts. One of several splice transcripts, VLT-ORF63a, encodes a unique protein; they also provide evidence in an in vitro neuronal culture system that this protein likely plays a direct role in the initiation of VZV gene transcription, thus driving the virus into a lytic state. This represents a very significant step forward in our understanding of how latency and reactivation are regulated by VZV. The work performed was meticulous and impressive. The manuscript is lucidly and concisely written. I have no recommendations for improving the manuscript.

We thank the reviewer for the very positive feedback.

Reviewer #2 (Remarks to the Author):

This study by Ouwendjik & Depledge et al demonstrates the existence of a novel fusion transcript in VZV (VLT-ORF63) that appears to be important for reactivation of the virus from latency. They use a number of complementary molecular biology techniques (dRNA-seq, cDNA seq, RT-qPCR and RACE), to clearly demonstrate and map the novel transcript during lytic replication and latent infection of a novel model system in addition to human TGs. Employing both RT-qPCR and dual RNA-FISH they demonstrate the existence of the transcript in natural latency (TGs isolated from human autopsy material). Interesting, the transcript appears to be absent from the in vitro model of latency. The authors produce antibodies that specially recognize the resultant protein to demonstrate its synthesis during lytic replication, but disappointingly do not use this antibody to probe latent samples nor following reactivation. Finally, they show that reactivation of VZV from in vitro infected differentiated human sensory neurons results in VLT-ORF63 transcription. In addition, transduction with VLT-ORF63 promotes reactivation from latency. In sum, the combined systems and approaches used generate a convincing study showing the presence of this novel transcript in VZV latency and/or reactivation. However, its role in reactivation could be strengthened, especially considering that these experiments do rely on a small number of experimental replicates.

We thank the reviewer for the positive words and constructive feedback, which we have used to improve our revised manuscript (as detailed below).

Major points.

1. A number of experiments seem to be underpowered and therefore lack rigor. There is no mention as to how power analysis was carried out. In particular for figures 4 and 5 using the differentiated human neurons, only 2 or 3 biological replicates have been carried out.

We have increased the number of biological replicates presented in Figure 4c (early reactivation; n=5), Figure 4d (anisomycin; n=6) and 4e (control; n=3) and Figure 5 (n=4 for each transduction). We show overall consistent results (e.g. L genes are only induced by VLT63-1, while induction of each genes within the same kinetic classes is variable).

2. Can the authors use their antibodies to investigate VLT-ORF63 protein synthesis in the human TGs or during reactivation in the in vitro model system? The localization of the mRNA in the

human TGs is interesting given its predominantly nuclear localization. The authors do not comment on this. Does this suggest that protein is not translated during latency?

Due to high levels of non-specific background staining in neuronal tissue both human TG and human iPSC-derived sensory neurons (HSN), neither the polyclonal chicken anti-pVLT-ORF63 antibody nor the polyclonal rabbit anti-pVLT antibody (Supplementary Figure 6) was available to detect endogenous pVLT-ORF63 even during reactivation in the in vitro model system. We have added the paragraph explaining this in the Discussion (lines 345-356).

3. The in vitro infected neurons undergo weak reactivation using stimuli such as neurotrophic factor deprivation, anisomycin or VTL-ORF63 transduction. Have the authors investigated whether a combination of anisomycin and VLT-ORF63 enhances reactivation?

Unfortunately, the limitation of our model system is that anisomycin, in addition to activation of JNK signaling, also inhibits protein synthesis (lines 321-325). Consequently, we cannot combine anisomycin and VLT-ORF63 to, potentially, further enhance reactivation, as pVLT-ORF63 cannot be produced. Similarly, protein synthesis of other viral proteins is blocked in cultures treated with anisomycin.

Minor points

1. Can the authors include the limit of detection information for the RT-qPCR primers used to be sure this is not differential expression of different transcripts in latency is not a detection issue?

We have now included more detailed information about the primer efficiency, melting temperatures and lower limit of detection for the primers sets used RT-qPCR (lines 485-491). Importantly, primer efficiency and sensitivity was similar for all primers, also shown in the standard curve analysis at the end of this response document.

2. More quantification of the FISH experiment would good showing the numbers of positive neurons for each probe, the numbers colocalizing verses present in the same neuron but distinct.

We agree with the reviewer that quantitative comparison of VLT- expressing versus VLT and ORF63 co-expressing neurons is of significant interest, also in relation to the presence of VZV DNA (see also question 2 from reviewer 3). The main purpose for the current experiments was to confirm that VLT and ORF63 ISH probes co-localize in latently VZV-infected neurons, indicative of VLT-ORF63 expression. While the pattern of VLT and VLT/ORF63 co-expression was consistent

between the donors, the number of VZV-positive and, especially total number of neurons, varied greatly between TG sections, thereby preventing accurate quantification in the current set of experiments. However, as detailed further in our response to question 2 from reviewer 3, such experiments are ongoing and part of our follow-up research.

3. Lns 199-211. This paragraph is confusing in the way that it is written. First “the same result” in In 202 is unclear given that two results are mentioned in the preceding sentence. Can they also comment more on the differential localization – it looks like following transfection ORF63 and VLT are mainly cytoplasmic proteins, whereas the VLT-ORF63 appears to be nuclear. However, during lytic it is nuclear and cytoplasmic.

We apologize for the confusion. We have now clarified these issues in the revised manuscript (now lines 219-231).

4. In figure 4c it’s not entirely clear what the black and white circles represent. Is this just one N for each or the means from the combined 40 samples? The text combined with the legend is confusing.

We apologize for the lack of clarity. We added 4 more samples for early/abortive reactivation shown in colored triangles (total 5 samples) and one fully reactivated sample is shown in white circle.

The text mentions that these are the samples that did not reactivate but the legend calls them early reactivation events. Perhaps these are abortive reactivations instead of early?

We revised sentence from “Although no infectious virus was detected in 38 replicates” in previous line 232 to “Although no infectious virus was recovered from 38 replicates” in the current line 252. We use term “complete reactivation” for reactivation with production of infectious progeny virus and revise “early reactivation” to “early/abortive reactivation” for reactivation with viral gene transcription other than VLT without detectable production of infectious progeny virus.

5. Can the authors do a direct comparison between IE, E and L transcripts in the differently transduced samples?

We thank the reviewer for the suggestion. Indeed, this way of presenting the data is more clear and we have reconstructed Figure 5 accordingly.

Does VLT-ORF63 specifically promote the viral DNA replication and/or transition to late gene expression?

In preliminary experiments, we did not detect robust viral DNA replication following VLT-ORF63 overexpression in latently VZV-infected HSN. However, it is very likely that our analysis did not have sufficient sensitivity to measure low-level VZV DNA replication.

Reviewer #3 (Remarks to the Author):

This is an important paper that reveals the existence of novel fusion transcripts, which combine the VZV latency-associated transcript (VLT) with the virus immediate early ORF63 gene. The fusion transcript (VLT-ORF63) is expressed during both lytic and latent infection and is shown to be expressed in human trigeminal ganglia. The authors demonstrate that one of the two major VLT-ORF63 isoforms encodes a novel fusion protein (pVLT-ORF63) and the functional significance of this novel protein is investigated in vitro using iPSC derived human sensory neuronal cell cultures. Using this system it is shown that whilst VLT is transcribed during latency, VLT-ORF63 can be induced by the application of reactivation stimuli. Subsequent studies reveal that expression of pVLT-ORF63 induced transcription of VZV lytic gene expression. Although full reactivation resulting in the production of infectious virus progeny was not observed in this system, the ability of pVLT-ORF63 to reverse transcriptional silencing of the latent VZV genome supports the authors view that expression of a novel VLT-ORF63 fusion transcript is likely to play a role in the transition from latency to lytic infection.

I have only a few comments.

We thank the reviewer for the supportive comments.

1. The detection of VLT-ORF63 transcripts in human trigeminal ganglia adds to the significance of paper but further explanation and consideration of the results are required. In Figure 2a it is not clear why there is a negative RT-qPCR signal for VL Texon A-1 primer set. This would rule out detection of Lytic VLT, Lytic VLT-ORF63a and Lytic VLT-ORF63b transcripts (supplementary Figure 1). This left me confused. Which VLT-ORF63 fusion transcripts are detected in human ganglia? I appreciate that the situation is clarified in part by the inclusion of 5'RACE data but it would be helpful if the characteristics of the lytic and latent VLT transcripts were discussed earlier in the results section.

We apologize for the lack of clarity. We add the sentence “By far the most common ^{lyt}VLT variants utilized a single upstream exon designated as exon A (Fig. 1a)” in lines 115-116, and also revised the sentence in previous lines 167-169 (now lines 183-186).

2. Figure 2b. The in situ hybridisation data on sections of human trigeminal ganglia confirmed expression of VLT or co-expressed nuclear VLT and ORF63 RNA. It would be helpful if the authors could clarify the proportion of neurons expressing VLT alone and give some indication whether VLT is expressed in all latent DNA containing cells. Is VLT a solid marker for latently infected cells? If the functional data presented favours the view that VLT-ORF63 fusion transcripts encode a function necessary for virus reactivation, is the detection of neurons expressing co-localizing ORF63/VLT in situ signal revealing cells at early stages of reactivation? Do these neurones express lytic gene transcripts?

We agree with the reviewer that it is important to investigate whether VLT-ORF63 is expressed by latently VZV-infected neurons or neurons undergoing VZV reactivation. To address this, we have performed additional mFISH experiments to stain for lytic gene ORF9, the most abundantly expressed VZV gene during lytic infection, in addition to VLT and ORF63 (new Supplementary Figure 3). Our data indicate that ORF9 is not expressed in human TG neurons, suggesting that VLT-ORF63 is indeed expressed by latently-infected neurons. We are optimizing protocols to combine DNA ISH with our RNA FISH methods to further detail patterns of viral RNA expression in latently VZV-infected human TG neurons, including a more complete quantification of VLT and/or VLT-ORF63 positive neurons.

3. The VZV sensory neuronal cell culture system provides a useful model to study VZV latency and reactivation and a key feature of this system is the establishment of a form of tight latency in which VLT is the sole transcript expressed. This appears to differ from the results presented for human trigeminal ganglia where the expression of both VLT and VLT-ORF63 fusion transcripts are commonly detected (compare Figure 2a with Figure 4a). Nonetheless, application of a reactivation stimulus by depletion of neurotropic factors resulted in the detection of multiple viral transcripts including VLT-ORF63a. Anisomycin treatment of cultures also induced expression of VLT and VLT-ORF63 isoforms but with more limited induction of lytic gene transcription. In order to determine whether induced transcription of VLT-ORF63 transcripts is directly involved in genome de-repression latently infected cultures were transduced with Lentivirus vectors expressing either ORF63 or the latent VLT-ORF63a or b isoforms. Both ORF63 and VLT-ORF63b

transduced cells resulted in sporadic IE and early gene activation whilst transduction with VLT-ORF63a resulted in the activation of all gene classes. In no instance was full virus reactivation observed. This result has tempered the conclusions, which proposes a potential role for the pVLT-ORF63 fusion protein in the transition from latency to reactivation. Given that the VLT-ORF63 isoforms are detected during latency in human trigeminal neurones, it may be premature at this stage rule out other potential functions of novel VLT-ORF63 isoforms that may relate to the known anti-apoptotic functions of ORF63 (Hood et al 2005., Gerada et al., 2018). Such a neuronal survival mechanism may be important both for stable maintenance of VZV latency and to facilitate efficient reactivation.

We agree with the reviewer that VLT-ORF63 may play additional roles, other than stimulating lytic VZV gene expression and have addressed these issues in the revised Discussion lines 338-341.

Reviewer comments, second round -

Reviewer #2 (Remarks to the Author):

The authors have addressed all comments.

Reviewer #3 (Remarks to the Author):

This is an important and convincing paper that clarifies the nature of latency associated transcripts expressed in neurones latently infected with VZV and reveals a key function of one spliced transcript encoding VLT ORF63a in reactivation. This revised manuscript and the inclusion of updated nomenclature is much improved and the authors have addressed the issues raised in my original review. This is a landmark paper which greatly advances our understanding of VZV latency and the nature of VZV transcription which is likely to have a key role in the transition from latency to lytic replication.